# Whole-Genome Sequencing and Genome Annotation of Pathogenic *Elsinoë batatas* Causing Stem and Foliage Scab Disease in Sweet Potato

**DOI:** 10.3390/jof10120882

**Published:** 2024-12-18

**Authors:** Yuan Xu, Yuqing Liu, Yihan Wang, Yi Liu, Guopeng Zhu

**Affiliations:** 1Sanya Nanfan Research Institute, Hainan University, Sanya 572025, China; 22210902000008@hainanu.edu.cn (Y.X.); 20213007007@hainanu.edu.cn (Y.L.); 23210902000002@hainanu.edu.cn (Y.W.); 2Key Laboratory for Quality Regulation of Tropical Horticultural Crops of Hainan Province, School of Tropical Agriculture and Forestry, Hainan University, Haikou 570228, China

**Keywords:** sweet potato, *Elsinoë batatas*, whole-genome sequencing, fungal pathogens

## Abstract

A pathogen strain responsible for sweet potato stem and foliage scab disease was isolated from sweet potato stems. Through a phylogenetic analysis based on the rDNA internal transcribed spacer (ITS) region, combined with morphological methods, the isolated strain was identified as *Elsinoë batatas.* To comprehensively analyze the pathogenicity of the isolated strain from a genetic perspective, the whole-genome sequencing of *E. batatas* HD-1 was performed using both the PacBio and Illumina platforms. The genome of *E. batatas* HD-1 is about 26.31 Mb long in 167 scaffolds, with a GC content of 50.81%, and 7898 protein-coding genes, 131 non-coding RNAs, and 1954 interspersed repetitive sequences were predicted. Functional annotation revealed that 408 genes encode virulence factors involved in plant disease (DFVF—Plant). Notably, twenty-eight of these virulence genes encode secretory carbohydrate-active enzymes (CAZymes), including two endo-1,4-β-xylanase genes and seven cutinase genes, which suggested that endo-1,4-β-xylanase and cutinase play a vital role in the pathogenicity of *E. batatas* HD-1 within sweet potato. In total, twelve effectors were identified, including five LysM effectors and two CDIP effectors, suggesting that LysM and CDIP effectors play significant roles in the interaction between *E. batatas* HD-1 and sweet potato. Additionally, our analysis of biosynthetic gene clusters (BGCs) showed that two gene clusters are involved in melanin and choline metabolism. This study enriches the genomic resources of *E. batatas* and provides a theoretical foundation for future investigations into the pathogenic mechanisms of its infection in sweet potatoes, as well as potential targets for disease control.

## 1. Introduction

The stem and foliage scab of sweet potato, caused by *Elsinoë batatas*, belongs to the phylum Ascomycota, class Dothideomycetes, order Myriangiales, family Elsinoaceae, and genus *Elsinoë*, and the disease primarily affects the leaves, vines, and tender shoots of sweet potato [1]. The veins on the underside of young leaves are particularly susceptible to infection. Typically, the disease leads to a yield reduction of 5% to 20%, and in severe cases, losses can exceed 50% [2,3,4]. Consequently, it poses a significant threat to the production and quality of edible sweet potatoes.

The disease has a long history of prevalence and is distributed worldwide, including in China, the United States, Brazil, Japan, Malaysia, and Uganda [5,6]. The optimum temperature for the spore germination of the scab pathogen is between 20 °C and 25 °C, and high humidity promotes disease development [7]. In China, it is widely found in coastal areas of the East China Sea, such as Guangdong, Guangxi, Fujian, Zhejiang, Taiwan, and Hainan [8,9]. These provinces’ high temperatures and humidity provide favorable conditions for pathogen invasion, particularly after summer typhoons.

Currently, eight genomic resources are available for *Elsinoë* species, including *E. ampelina* YL-1 [10], *E. fawcettii* SM-16 and *E. fawcettii* DAR-70024 [11], *E. fawcettii* BRIP53147a [12], *E. australi* Ea-1 [11], *E. australi* NL1 [13], *E. murrayae* [14], and *E. batatas* CRI-CJ2 [7]. In this study, we isolated a new pathogenic strain and sequenced its whole genome using second- and third-generation sequencing technologies. Furthermore, we conducted a bioinformatics analysis based on the whole-genome sequence, which is crucial for disease prevention and control, as well as for understanding plant–pathogen interactions.

## 2. Materials and Methods

### 2.1. Strain Isolation and Culture

In naturally infected fields, typical diseased plants of the sweet potato variety “Fucaishu 1830-11” were randomly collected (Figure 1). Fresh sweet potato petioles and stems exhibiting characteristic red lesions were selected for further analysis. Tissue blocks, approximately 0.3 cm in size, were cut at the boundary between healthy and diseased tissue via sterilized scissors within a laminar flow hood (SW-CJ-2D, SWCJ, Suzhou, China). The tissue blocks were initially disinfected with 75% ethanol for 35 s, rinsed once with sterile water, then treated with 2% sodium hypochlorite for 45 s, and finally rinsed three times with sterile water. Excess moisture was carefully removed using sterile absorbent paper. The tissue blocks, with the lesion side facing down, were placed on potato dextrose agar (PDA, Coolaber, Beijing, China) medium supplemented with 0.1% L-lactic acid (Coolaber, Beijing, China) and 100 μg/mL streptomycin (Coolaber, Beijing, China). The plates were sealed with Parafilm and incubated under light conditions at 28 °C in a biochemical incubator (SPX-360BE, Shanghai LICHEN-BX Instrument Technology Company, Shanghai, China). After 4 d, the growth of the pathogen was observed and recorded. Tissue blocks that showed no contamination were then transferred to fresh PDA plates containing 0.1% L-lactic acid and 100 μg/mL streptomycin using sterile tweezers. This sample was incubated at 28 °C for 3 days. When red-brown colonies appeared at the interface between the tissue blocks and the medium, the colony edge was carefully picked using a sterile pipette tip and inoculated onto fresh PDA medium. Single colonies were isolated through repeated streaking until consistent colony morphology was observed on the PDA medium, resulting in the purified pathogen strain HD-1. This rigorous process ensured the isolation and purification of the pathogen, facilitating further studies on its pathogenicity and molecular characteristics.

### 2.2. Morphological Observation of the Strain

Fungal blocks cultured on PDA medium for 30 days were selected. A fungal cake from the edge of the colony was transferred to a sterile centrifuge tube containing 10 mL of sterile water and vortexed for 10 min. Subsequently, 200 μL of the fungal suspension was spread onto a new PDA plate and incubated at 28 °C for 7 days. After incubation, 2–3 mL of sterile water was added to the plate, and the colony surface was scraped repeatedly with a sterile scalpel. The resulting conidial suspension was then filtered through three layers of sterile lens paper. Under an optical microscope, five random fields of view (400× magnification) were selected for observation. The conidial structure and size were measured for each field.

### 2.3. Strain Identification

Representative mycelia of strain HD-1 cultured on potato dextrose agar (PDA) for 7 d were collected. The fungal blocks were cut using a sterile scalpel and ground in liquid nitrogen, and genomic DNA was extracted via the Rapid Fungi Genomic DNA Isolation Kit (Sangon Biotech, Shanghai, China). The genomic DNA was stored at −20 °C for future use. PCR amplification of the ITS region was performed via the primer pair ITS1 (5′-TCCGTAGGTGAACCTGCGC-3′) and ITS4 (5′-TCCTCCGCTTATTGATATGC-3′), synthesized by Sangon Biotech (Shanghai, China). The 25 μL PCR reaction mixture consisted of 12.5 μL of 2 × TransTaq @ HiFi PCR SuperMix (TransGen Biotech, Beijing, China), 1 μL of each primer (10 μmol/L), 9.5 μL of nuclease-free water, and 1 μL of the DNA template. The PCR program included predenaturation at 94 °C for 1 min, denaturation at 94 °C for 1 min, annealing at 55 °C for 50 s, extension at 72 °C for 1 min for a total of 35 cycles, and a final extension at 72 °C for 10 min, which was amplified in a PCR machine (RePure-A, Hangzhou Bio-Gener Technology Company, Hangzhou, China). The PCR products were subjected to 1% agarose gel (Coolaber, Beijing, China) electrophoresis and sent to Hainan NanShan Biotechnology Co., Ltd., for sequencing. The obtained sequences were submitted to NCBI to construct a sequence alignment of various fungal ITS regions. A phylogenetic tree was constructed via MEGA 11.0 with the neighbor-joining method and 1000 bootstrap replicates to estimate branch support.

### 2.4. Pathogenicity Assay of Strain

To identify strain HD-1 as the pathogen of sweet potato stem scab, we conducted a pathogenicity test according to Koch’s postulates [15]. A representative colony of strain HD-1 cultured on PDA medium was selected, and an 8 mm fungal disk was taken from the colony’s edge and transferred to 100 mL of PDB liquid medium (Coolaber, Beijing, China). The culture was incubated in a constant-temperature shaker (ZQTY-70V, Shanghai Zhichu Instrument Company, Shanghai, China) at 200 rpm and 28 °C for 30 days. Subsequently, 100 μL of the fungal suspension was spread onto a new PDA plate and incubated under light conditions at 28 °C for 7 days. After incubation, 1–2 mL of sterile water was added to the plate, and the edge of the colony surface was scraped repeatedly with a sterile scalpel to dissolve the spores in the water. The suspension was filtered through three layers of sterile lens paper to obtain a conidial suspension. A hemocytometer was then used to measure the conidial concentration, and a PDA medium was utilized to adjust the concentration to 3.4 × 10^6^ conidia/mL. Tween 80 (Coolaber, Beijing, China) was added to achieve a final concentration of 2% in the conidial suspension.

The “Fucaishu 1830-11” seedlings were planted in sterilized nutrient soil and grown in a plant growth incubator (28 °C, 8 h light/16 h dark). The conidial suspension was evenly sprayed onto young leaves, shoots, and stems using a sprayer. Five plants were inoculated, ensuring that each plant was sprayed until the liquid dripped off. Plants inoculated with sterile water served as controls. The inoculated plants were placed in a plastic greenhouse outdoors, where the temperature during the inoculation period ranged from 20 °C to 28 °C, and water was regularly sprayed to maintain a relative humidity of ≥75%. Disease symptoms in the sweet potato plants were observed and recorded daily. The experiment was repeated three times.

### 2.5. DNA Preparation

*E. batatas* was grown on potato dextrose agar (PDA) at 28 °C for 7 d, and mycelial plugs were then transferred to a 100 mL flask containing 60 mL of potato dextrose broth (PDB), where they were incubated at 28 °C and 200 rpm for 7 d for mycelium collection for DNA extraction. The genomic DNA of *E. batatas* was extracted via the Genomic DNA Purification Kit (A1120, Promega, Madison, WI, USA). High-quality DNA (OD260/280 = 1.8–2.0, >10 µg; NanoDrop2500, Thermo Fisher Scientific, Waltham, MA, USA) was used for sequencing.

### 2.6. Library Construction and Sequencing

The genome was sequenced via the Illumina and PacBio_SMRT sequencing platforms.

For Illumina sequencing, genomic DNA from the strain HD-1 was used for sequencing library construction. The DNA samples were sheared into ~400 bp fragments via a Covaris M220 focused acoustic shearer following the manufacturer’s protocol. Illumina sequencing libraries were prepared from the sheared fragments via the NEXTFLEX^®^ Rapid DNA-Seq Kit (Revvity, Waltham, MA, USA). Briefly, the 5′ ends were first end-repaired and phosphorylated. Next, the 3′ ends were A-tailed and ligated to sequencing adapters. The third step involved enriching the adapter-ligated products using PCR. The prepared libraries were then used for paired-end Illumina sequencing on an Illumina NovaSeq X Plus (Illumina Inc., San Diego, CA, USA).

For PacBio sequencing, DNA samples were sheared into 8–10 kb fragments via G-tubes. Each end of the single-stranded DNA was subsequently connected to a double-stranded sequence of positive and negative chains. The PacBio library was subsequently prepared and sequenced on the PacBio_SMRT platform.

The genome of *E. batatas* HD-1 has been deposited at GenBank under BioProject PRJNA1156576.

### 2.7. Genome Assembly and Annotation

The data generated from Illumina and PacBio_SMRT sequencing platforms were used for bioinformatics analysis. All of the analyses were performed using the free online Majorbio Cloud Platform (http://cloud.majorbio.com, accessed on 25 May 2022) from Shanghai Majorbio Bio-pharm Technology Co., Ltd. (Shanghai, China). The detailed procedures are as follows.

The raw Illumina sequencing reads generated from the paired-end library were subjected to quality filtering using fastp v0.23.0 with a Q-score >= 20 and a read length >= 15, by which the low-quality data can be removed to form clean data. PacBio reads were extracted, base-called, demultiplexed, and trimmed with a minimum Q score cutoff of 7. The PacBio reads were assembled into scaffolds via Flye version 2.9.2 (https://github.com/mikolmogorov/Flye, accessed on 18 March 2023). Then, we used SOAP2 software to map the clean data of Illumina sequencing onto the assembly sequence to obtain the GC_depth to determine whether there was contamination of other genomes in the sample *E. batatas* HD-1. Finally, BUSCO (Benchmarking Universal Single-Copy Orthologs, Version 5.4.5) was utilized to evaluate the integrity of the genome assembly.

The prediction of open reading frames (ORFs) was performed via Maker2, while tRNAs were predicted via tRNAscan-SE v2.0.12, and rRNA was identified via Barrnap v0.9. The predicted ORFs were annotated through various databases, including NR, Swiss-Prot, Pfam, GO, COG, KEGG, and CAZY. The GO annotation was performed by blast2go. The NR, Swiss-Prot, COG, and KEGG annotation was performed by Diamond. The CAZY annotation was performed by Hmmscan and Diamond. Briefly, each set of query proteins was aligned with the databases, and annotations of the best-matched subjects (E-value < 10^−5^) were obtained for gene annotation. The prediction of biosynthetic gene clusters (BGCs) was performed via the antiSMASH v6.1.1 tool [12].

The prediction and annotation of pathogenicity-related genes were conducted via the pathogen–host interaction (PHI) database (http://www.phi-base.org/, accessed on 1 July 2024) and the Database of Fungal Virulence Factors (DFVF) (http://sysbio.unl.edu/DFVF/index.php, accessed on 18 March 2023). The analysis was performed with Diamond software (v0.8.35) (http://www.phi-base.org/, accessed on 1 July 2024), applying an E-value cutoff of 1 × 10^−5^. The screening criteria involved multiple rounds of filtering, retaining only the best match for each gene.

Secreted proteins were analyzed using Diamond software. SignalP was utilized to identify signal peptide sites (http://www.cbs.dtu.dk/services/SignalP/, accessed on 1 June 2024), while TMHMM (v2.0) was employed to predict transmembrane helix structures (http://www.cbs.dtu.dk/services/TMHMM/, accessed on 1 June 2024).

### 2.8. Genome Synteny Analysis

The genome synteny analyses between the *E. batatas* strain CRI-CJ2 and HD-1 genome were performed by using the free online Majorbio Cloud Platform from Shanghai Majorbio Bio-pharm Technology Co., Ltd. (Shanghai, China). Sebelia was used for synteny analysis, and Circos software was used to draw a synteny cyclic graph.

## 3. Results

### 3.1. Morphological and Phylogenetic Tree of the Isolated Strain

The representative strain HD-1, isolated and purified from sweet potato scab disease samples, was selected for identification. The strain was cultured on PDA at 28℃, and its morphological characteristics were observed. After 30 days, the colonies appeared nearly circular with irregular edges and presented a red color. The colony surface was covered with varying amounts of white, woolly aerial mycelium, and its diameter ranged from 1.7 to 2.6 cm. The reverse side of the colony exhibited brownish-red pigmentation. As the colony matured, it developed a raised, hill-like center, with wrinkles along the edges. The mycelium was densely packed, forming bundles or fence-like layers (Figure 2a,b). Under optical microscopy, the conidia of strain HD-1 were observed to be unicellular, hyaline, and non-septate and ranged from elliptical to fusiform in shape, measuring 6.16–10.02 µm × 2.79–3.38 µm (Figure 2c). Based on these morphological characteristics, strain HD-1 was preliminarily identified as *E. batatas*.

The rDNA-ITS sequence of strain HD-1 was amplified using the universal primers ITS1 and ITS4. Bidirectional sequencing revealed that the rDNA-ITS sequence was 601 base pairs (bps) long. Phylogenetic analysis of this sequence indicated that strain HD-1 clustered together with the *Elsinoë batatas* strain SPEb-1 (GenBank accession number MN266887), with 96% bootstrap support (Figure 3). Given the morphological characteristics and the results of the phylogenetic analysis, the strain HD-1 was conclusively identified as *E. batatas.*

### 3.2. Pathogenicity Test of the Pathogen on an Edible Sweet Potato

A conidial suspension of strain HD-1 was inoculated onto the edible sweet potato variety “Fucaishu 1830-11”. After 10 days, small, light brownish-red lesions and pinhead-sized lesions appeared on the petioles and the stem leaf veins, and the surfaces of the petioles and stems began to wrinkle (Figure 4). The young leaves at the apex exhibited deformation, taking on a “bent-knee” appearance. These symptoms closely matched those observed in naturally infected field plants. Pathogenic fungi were re-isolated from diseased leaves, and their characteristics matched those of the original strain, confirming that strain HD-1 is indeed the causative agent of sweet potato scab.

### 3.3. Genomic Characteristics and Assembly

The genome of *E. batatas* was sequenced using the PacBio and Illumina platforms. About 6.04 G bases and 9.44 G bases were generated via Illumina and PacBio_SMRT sequencing, respectively. A total of 167 scaffolds were assembled, resulting in a genome size of 26.31 Mb with a GC content of 50.81%. The N50 and N90 fragment lengths were 1.05 Mb and 0.33 Mb, respectively. The predicted protein-coding genome spanned 16.65 Mb and contained 7898 coding genes, accounting for 63.28% of the genome. Additionally, 131 non-coding RNAs (ncRNAs) were identified, including 26 ribosomal RNAs (rRNAs) and 105 transfer RNAs (tRNAs). The genome also contained 4.00 Mb of repeat sequences, accounting for 15.19% of the total genome. The interspersed repeats included 1153 long terminal repeats (LTRs), 8 short interspersed nuclear elements (SINEs), 230 long interspersed nuclear elements (LINEs), and 563 DNA transposons. The detailed assembly statistics are summarized in Table 1. A circular graphical map of the whole *E. batatas* HD-1 genome created using the Circos software (v 0.69-8) is presented in Figure 5. From the center to the outer circle, the first circle represents the GC skew, the second represents the GC content, the third represents CAZymes, and the fourth circle represents the scale. This comprehensive genomic analysis provides a detailed understanding of the genetic composition and structure of *E. batatas* strain HD-1, which can be valuable for further research and control strategies for sweet potato scab disease.

In this study, GC_depth was used to evaluate whether there is contamination of other genomes in *E. batatas* HD-1 (Figure 6). It indicated that there is no contamination in the *E. batatas* HD-1 genome. In addition, BUSCO software (https://gitlab.com/ezlab/busco/-/releases/5.4.5/, accessed on 30 January 2023.) was used to evaluate the integrity and completeness of the genome assembly. This software assesses the quality of genome assembly based on the evolutionary information of single-copy homologous genes that are conserved across species within the fungal community [16]. The BUSCO results indicated that 93.3% of the single-copy genes were fully aligned with the fungal reference set, which was marginally lower than the 97% BUSCO completeness observed in CRI-CJ2 [7]. This indicates a high level of genome completeness and assembly quality. This assessment provides strong evidence that the genome assembly is of high quality and reliability, which is crucial for further downstream analyses and functional studies of *E. batatas* HD-1.

### 3.4. Annotation of the E. batatas HD-1 Gene

To better understand the *E. batatas* HD-1 genome, functional annotations were performed using various databases, including NR, COG, GO, and KEGG. Among the 7898 coding genes, 100% were annotated in the NR database, while 68.54% and 71.70% were annotated in Swiss-Prot and Pfam, respectively. Detailed annotation information is provided in Appendix A.

COG analysis classified 2996 genes into functional categories (Figure 7). The largest group (465 genes) was involved in carbohydrate transport and metabolism (Class G), followed by lipid transport and metabolism (Class I, 309 genes) and amino acid transport and metabolism (Class E, 282 genes) (Figure 6).

The GO annotation revealed 5568 genes (70.51%) in three categories: molecular function (4327 genes), cellular component (3312 genes), and biological process (3013 genes). As shown in Figure 8, the genes were enriched mainly in “catalytic activity” (2903 genes), “cellular anatomical entity” (2900 genes), “binding” (2336 genes), “metabolic process” (2241 genes), and “cellular process” (2153 genes).

The KEGG annotation showed that 5203 (74.04%) genes were categorized into six major classes, including 48 subcategories. The class with the greatest enrichment was “metabolism” (3689 genes); the class with the second greatest number of genes was “human disease” (1362 genes), followed by “genetic information processing” (969 genes) (Figure 9). In total, 218 genes were linked to secondary metabolites, of which 64 were involved in the metabolism of terpenoids and polyketides. Further analysis of the combined DFVF—Plant revealed that three genes encoding DFVF—Plant were involved in terpenoid metabolism, including STE24 endopeptidase (gene4914) and acetyl-CoA C-acetyltransferase (gene2205 and gene5168).

### 3.5. Analysis of CAZymes

CAZymes organize the catalytic and substrate-binding modules of proteins involved in assembling and breaking down complex carbohydrates into sequence-based families [17]. A total of 333 genes encoding CAZymes were identified using the CAZy database (Appendix A), accounting for 4.83% of the total gene count in the *E. batatas* genome. These genes are categorized into five protein families (Table 2): glycoside hydrolases (GHs), auxiliary activities (AAs), carbohydrate esterases (CEs), polysaccharide lyases (PLs), and glycosyl transferases (GTs).

GHs, which catalyze the hydrolytic cleavage of glycosidic bonds to generate carbohydrate hemiacetals, constitute the largest family with 160 genes. The majority of these genes are distributed among the GH43 (sixteen genes), GH16 (thirteen genes), and GH5 (nine genes) families. Fifty-nine genes encode AAs, which are redox enzymes that act in conjunction with CAZymes and are found primarily in the AA3 (nineteen genes), AA9 (ten genes), AA7 (seven genes), and AA1 (seven genes) families. Additionally, there are fifty-one genes encoding CEs, mainly concentrated in the CE10 (seventeen genes), CE1 (ten genes), and CE5 (nine genes) families, which catalyze the de-O-acetylation or de-N-acylation of substituted saccharides. The numbers of glycosyl transferases (GTs) and polysaccharide lyases (PLs) are 51 and 12, respectively.

### 3.6. Analysis of Virulence-Factor-Related Genes

Virulence-factor-related genes were predicted using the fungal virulence factor (DFVF) database, and a total of 892 (11.29%) genes were identified as associated with the disease (as shown in Appendix A). Among these genes, 408 genes were implicated in plant disease, while 484 genes were involved in animal or human disease. The majority of the genes associated with plant disease were related to leaf spot (134 genes), followed by rice blast (64 genes), gray mold (49 genes), and smut (49 genes) (Figure 10a).

The GO functional annotations of the genes encoding virulence factors associated with DFVF—Plant were also analyzed (Figure 10b). The genes were categorized into “molecular function” (353 genes), “biological process” (103 genes), and “cellular component” (306 genes). The genes were predominantly involved in “catalytic activity” (281 genes), “binding” (222 genes), “metabolic process” (176 genes), and “cellular process” (197 genes).

The secretory CAZymes associated with plant pathogenicity were screened by comparing the gene profiles of the secretome, CAZymes, and DFVF—Plant. The predicted secretome revealed a total of 662 genes encoding secretory proteins (as shown in Appendix A). A Venn diagram indicated that 28 genes were common among the secretome, CAZymes, and DFVF—Plant (Figure 11a). The 28 secretory CAZymes associated with plant pathogenicity belong primarily to the GH, CE, AA, and PL families (Figure 11b). The GO functional analysis revealed that these 28 secretory CAZymes are crucial for catalytic activity, metabolic processes, binding, and cellular processes (Figure 11c). Among the twenty-eight secretory CAZyme genes, two genes encode endo-1,4-β-xylanases, which are involved in xylan degradation, and seven genes encode cutinases that catalyze the hydrolysis of cutin.

### 3.7. Analysis of PHI

A whole-proteome BLAST search against the pathogen–host interaction (PHI) database identified a total of 2032 (25.7%) genes, which were primarily categorized into eight groups, as shown in Appendix A. As illustrated in Figure 12, 82 genes were associated with hypervirulence, 12 were linked to effectors, and 117 genes were related to lethal. The remaining genes were distributed across chemical resistance (4), unaffected pathogenicity (843 genes), loss of pathogenicity (210 genes), enhanced antagonism (1 gene), and reduced virulence (988 genes). Among the twelve effectors identified, five are LysM effectors (gene3300, gene1743, gene6443, gene6005, and gene5765), and two are CDIP effectors (gene7379 and gene6212).

### 3.8. Prediction of Secondary Metabolite Biosynthetic Gene Clusters

Secondary metabolites are primarily classified into four chemical families: nonribosomal peptides (NRPs), polyketides (PKs), NRP/PKS hybrids, and terpenoids [18]. The genes encoding secondary metabolites tend to aggregate in biosynthetic gene clusters (BGCs) [19]. The analysis of secondary metabolite gene clusters using the AntiSMASH tool identified a total of 15 gene clusters, which were categorized into five types (Figure 13), including terpenes, which had the highest number of gene clusters (n = 5), followed by type I polyketide synthase (T1PKS) (n = 4), nonribosomal peptide synthetase (NRPS) (n = 4), type III polyketide synthase (T3PKS) (n = 1), and INDOLE/T1PKS/NRPS clusters (n = 1). Remarkably, two BGCs exhibited a 100% similarity to previously documented clusters, including choline and 1,3,6,8-tetrahydroxynaphthalene. The latter is an indispensable precursor of DHN melanin, which serves as the precursor molecule of most fungal melanins [20].

### 3.9. Comparative Genome Analysis of E. batatas

Genome synteny analysis showed that 99.87% of the *E. batatas* HD-1 genome sequence was aligned to the *E. batatas* CRI-CJ2 genome. There is a particularly high degree of synteny between the genomes of the strains HD-1 and CRI-CJ2, suggesting a highly similar genetic background between the two strains (Figure 14).

## 4. Discussion

In this study, we used the ITS sequence to identify the species of the isolated strain. The phylogenetic tree analysis revealed that strain HD-1 clustered closely with *Elsinoë batatas* strain SPEb-1. These morphological characteristics of HD-1 were consistent with previous findings of Zhang et al. [7]. Based on both morphological similarities and phylogenetic evidence, the isolated strain HD-1 was identified as *E. batatas*. Furthermore, the pathogenicity test confirmed that strain HD-1 is capable of causing sweet potato scab disease. These findings reinforce the accuracy of the identification and its association with the disease in sweet potatoes.

To gain a comprehensive understanding of the gene distribution in *E. batatas* HD-1, we systematically annotated gene functions using multiple databases, including the NR, Swiss-Prot, GO, COG, KEGG, PHI, and CAZy databases. These annotations provide deeper insights into the pathogenic mechanisms of this strain in sweet potatoes. CAZymes are essential for fungal survival and pathogenicity, as they degrade host components, facilitating nutrient acquisition and the establishment of infection [21]. Therefore, predicting genes encoding CAZymes is crucial for understanding the pathogenesis of *E. batatas*. In strain HD-1, the GH and AA families contained the greatest number of CAZymes. GHs play a well-documented role in fungal nutrition, particularly in terms of plant biomass degradation and cell wall remodeling [22,23]. Specifically, GH16 and GH5 are linked to plant biomass degradation. GH5 shows strong degradative capabilities and is involved in cell wall remodeling [24,25,26]. Cell wall remodeling is critical for fungal survival and interaction with host tissues, allowing pathogens to establish infection [27]. The AA3 and AA9 families are associated with cellulose and hemicellulose degradation [28], while the AA7 family contains oligosaccharide oxidases that transfer electrons to lytic polysaccharide monooxygenases, enhancing cellulose degradation [29]. These findings suggest that *E. batatas* HD-1 possesses numerous CAZymes capable of degrading lignocellulose, enabling nutrient acquisition from host plants and enhancing its survival. This comprehensive understanding of CAZyme distribution and function provides critical insights into the pathogenic strategies employed by *E. batatas* HD-1 and may inform future strategies for disease management in sweet potatoes. The analysis of virulence factors using the DFVF revealed that 45.74% of the 892 virulence genes in *E. batatas* HD-1 are associated with plant diseases, with the majority being related to leaf spots. It has been suggested that some virulence genes involved in carbohydrate metabolism are linked to horizontal gene transfer, which could potentially lead to sudden outbreaks of highly virulent fungal strains [30]. This phenomenon underscores the dynamic and evolving nature of fungal pathogenicity.

Our comprehensive analysis identified 28 secretory CAZymes associated with DFVF—Plant, including endo-cutinase and 1,4-β-xylanase. These enzymes play critical roles in the degradation of plant structures that form barriers to infection. Fungal cutinase is an inducible enzyme that degrades the host cuticle, the first barrier that pathogens must overcome during invasion. This degradation facilitates pathogen entry and weakens host resistance [31]. For instance, the hemibiotrophic pathogen *P. brassicae* produces cutinase to directly penetrate the host cuticle, a process essential for the pathogenesis of brassicas caused by *P. brassicae* [32,33]. The plant cell wall, primarily composed of hemicellulose, serves as a key defense barrier against pathogens [34]. Endo-β-1,4-xylanase is involved in the degradation of plant cell walls, suggesting its vital role in pathogenesis [35]. In *Botrytis cinerea*, disruption of the endo-β-1,4-xylanase gene (*xyn11A*) significantly reduces virulence [36]. Similarly, in *Magnaporthe oryzae*, the virulence of the strain was severely impaired after the loss of the β-1,4-xylanase gene MoXYL1A, which likely promotes the virulence of *M. oryzae* by interfering with the proper function of the host chloroplast [37]. Based on existing research results, we suggest that endo-1,4-β-xylanase and cutinase are key factors in the pathogenicity of *E. batatas* HD-1 within sweet potato. These enzymes enable the fungus to degrade essential structural components of the host plant, facilitating its entry and colonization.

In addition to virulence factors, secondary metabolites play a critical role in fungal pathogenicity [38]. In this study, we identified three DFVF—Plant genes related to terpenoid metabolism which are known for their diverse biological activities [39]. Terpenoids can act as signaling molecules, toxins, or defense agents, making them crucial for the survival and pathogenicity of fungal species. Fungal melanins are pigments that protect fungi from environmental stresses and also serve as virulence factors, conferring pathogenicity [40,41]. Our analysis of BGCs revealed one T1PKS gene cluster involved in melanin production. Choline, a key component of the major membrane phospholipid phosphatidylcholine, is essential for fungal growth and hyphal morphology [42]. Choline metabolism is crucial for the structural integrity and function of fungal membranes, impacting the overall fitness and pathogenicity of the fungus. The identification of these gene clusters related to melanin and choline metabolism underscores the multifaceted nature of fungal pathogenicity. These secondary metabolites and their biosynthetic pathways provide the fungus with crucial adaptations that enhance its survival and virulence in the host environment.

Effectors are secreted pathogen proteins that play a critical role in interactions between pathogens and plants. These proteins are often targeted to either the host cytoplasm or apoplast, which enables the pathogen to evade the recognition receptor activities of the host’s defense system, thereby facilitating infection in the plant [43,44]. Effectors typically interact with host proteins to suppress or modulate host defenses, allowing the pathogen to establish and maintain infection. Chitin, a major component of fungal cell walls, acts as a microbe-associated molecular pattern (MAMP) that is recognized by plant receptors containing lysin motifs (LysMs); this recognition activates immune defenses [45,46]. In *Zymoseptoria tritici*, LysM effectors such as Mg1LysM, Mg3LysM, and Mgx1LysM have been shown to protect fungal hyphae from host chitinases, thereby disabling wheat’s chitin-triggered immunity [30]. Currently, LysM effectors have been identified in various fungi, including *Magnaporthe oryzae* (Slp1), *Colletotrichum higginsianum* (Elp1 and Elp2), *Verticillium dahliae* (Vd2LysM), *Rhizoctonia solani* (RsLysM and LysM1), and *Clonostachys rosea* (LysM2) [47,48,49,50,51]. Furthermore, in *Magnaporthe oryzae*, effectors such as *MoCDIP1*, *MoCDIP3*, *MoCDIP4*, *MoCDIP5*, *MoCDIP6*, *and MoCDIP1* have been shown to induce host cell death in rice [52,53]. Specifically, *MoCDIP4* binds to *OsDjA9* and prevents the degradation of *OsDRP1E*, resulting in reduced resistance in rice [54]. Through a whole-proteome BLAST search against the PHI database, we identified twelve effectors in *E. batatas* HD-1, including five LysM and two CDIP effectors. This suggests that LysM and CDIP effectors play significant roles in the interaction between *E. batatas* HD-1 and sweet potato. These effectors likely function to suppress sweet potato immunity, enabling the fungus to successfully infect and colonize the host plant. The identification of these effectors in *E. batatas* HD-1 highlights the importance of effector proteins in the interaction between pathogens and their host plants. In particular, LysM and CDIP effectors are crucial for evading host immunity and facilitating infection.

Previously, Zhang et al. published the genome of *E. batatas* CRI-CJ2 [7]. To explore the genetic background similarity of HD-1 and CRI-CJ2, we performed a synteny analysis of the genomes of the two strains. The result showed that the two strains share a very similar genetic makeup. The high similarity also indicates that the genetic material encoding functional genes, regulatory regions, and other genomic features is conserved between the two strains. This information is useful for comparative genomics studies, breeding programs, and understanding the genetic basis of specific traits or associated diseases.

The analysis of CAZyme distribution and function in *E. batatas* HD-1 sheds light on the pathogenic strategies used by this fungus against sweet potatoes. The findings emphasize the potential to develop management strategies for fungal diseases by focusing on key CAZymes vital for the pathogen’s virulence. Moreover, this study highlights the fungal secondary metabolites and effector proteins promoting fungal pathogenicity. These elements work together to subvert host immune defenses, facilitating successful infection and colonization. Overall, understanding these mechanisms enhances our knowledge of fungal pathogenicity, laying the foundation for more effective disease control in sweet potatoes.

## 5. Conclusions

Using tissue separation techniques, this study successfully isolated *E*. *batatas* strain HD-1, the causative agent of sweet potato scab disease. Whole-genome sequencing of *E. batatas* HD-1 provided critical insights into its genetic composition, identifying 7898 protein-coding genes, 131 non-coding RNAs, and 1954 repetitive elements. Functional analyses uncovered a diverse array of genes associated with pathogenicity, including 1182 hypervirulence genes, 117 lethal genes, and 12 effector genes, which contribute to a better understanding of the molecular mechanisms underlying its infection process. The identification of 333 CAZyme genes highlights the pathogen’s ability to degrade plant cell walls, facilitating host infection. Additionally, 15 secondary metabolite gene clusters were identified, including NRPS and PKS clusters, providing further insight into the pathways related to fungal virulence and pathogenicity. These findings suggest new potential targets for disease management strategies aimed at controlling sweet potato scab disease, ultimately contributing to improved agricultural practices and enhanced crop protection.

## Figures and Tables

**Figure 1 jof-10-00882-f001:**
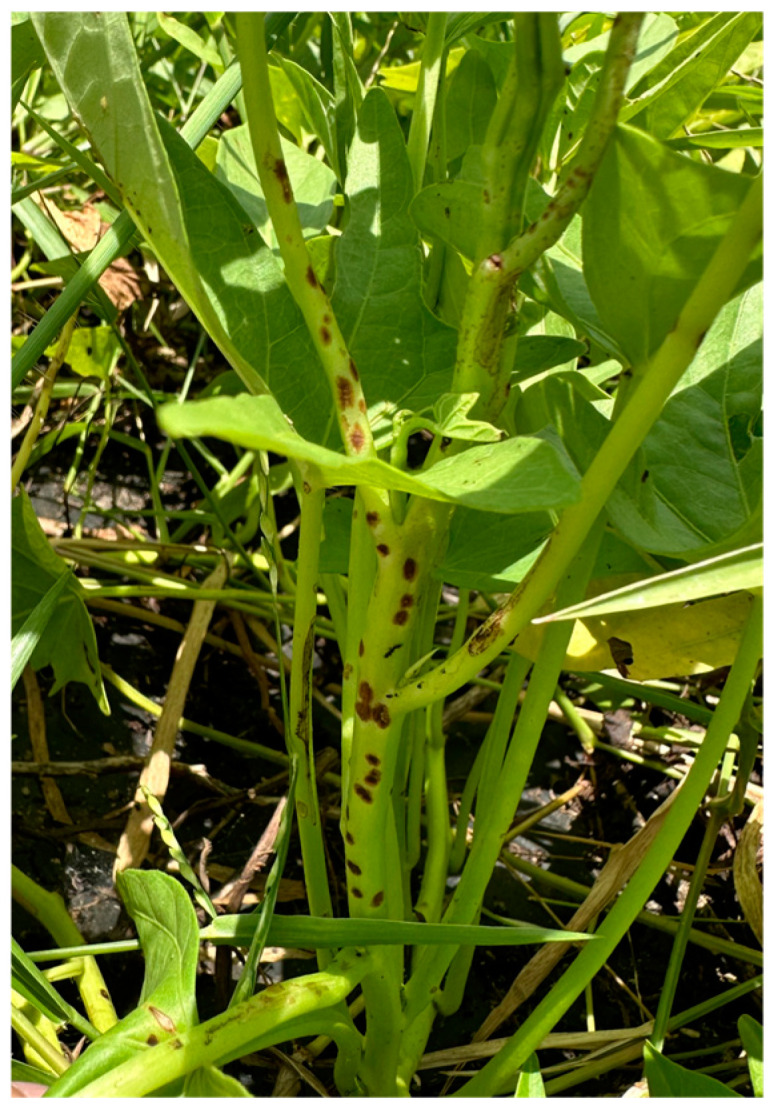
“Fucaishu 1830-11” typical diseased plant.

**Figure 2 jof-10-00882-f002:**
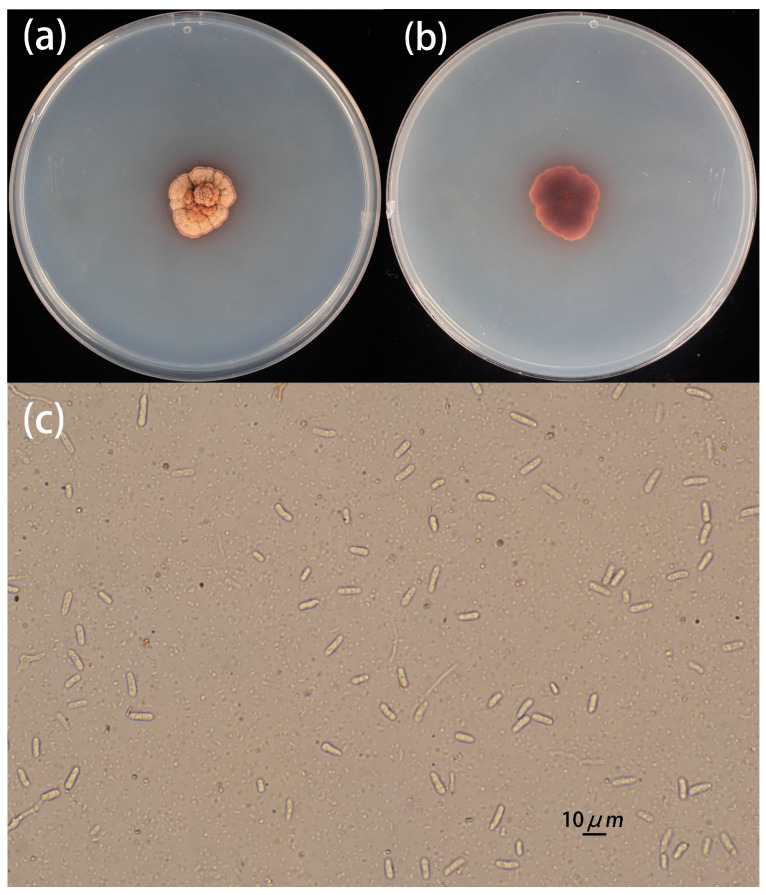
Morphology of filamentous fungus: (**a**) the front view of the colony morphological characteristics of strain HD-1 cultured for 30d; (**b**) the back view of the colony morphological characteristics of strain HD-1 cultured for 30d; and (**c**) conidia image (bar = 10 µm).

**Figure 3 jof-10-00882-f003:**
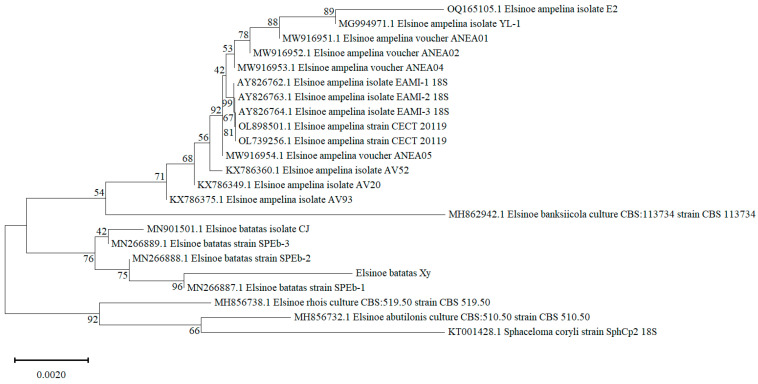
The neighbor-joining (NJ) phylogenetic tree was constructed based on ITS sequences.

**Figure 4 jof-10-00882-f004:**
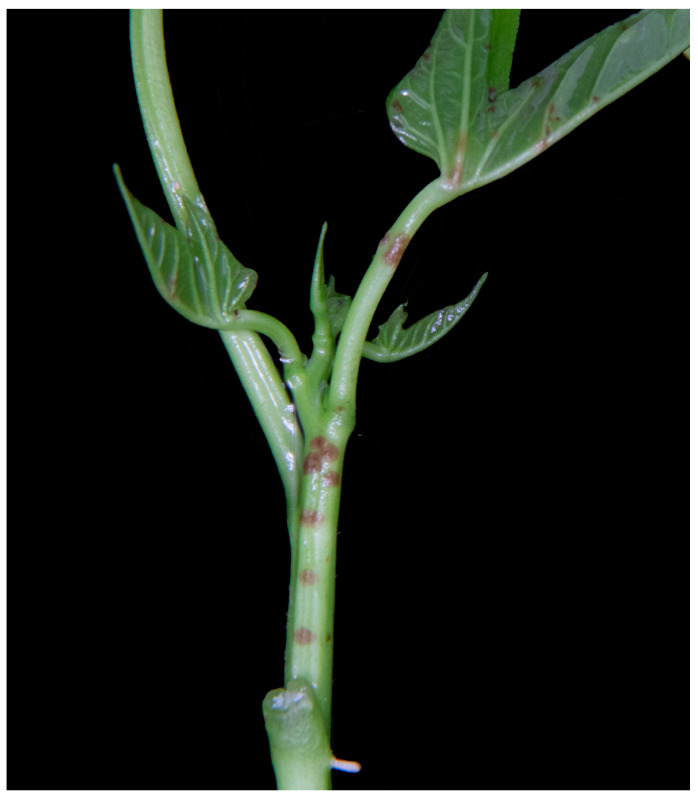
Symptoms of infection in sweet potato “Fucaishu1830-11” plants after 10 d of infection.

**Figure 5 jof-10-00882-f005:**
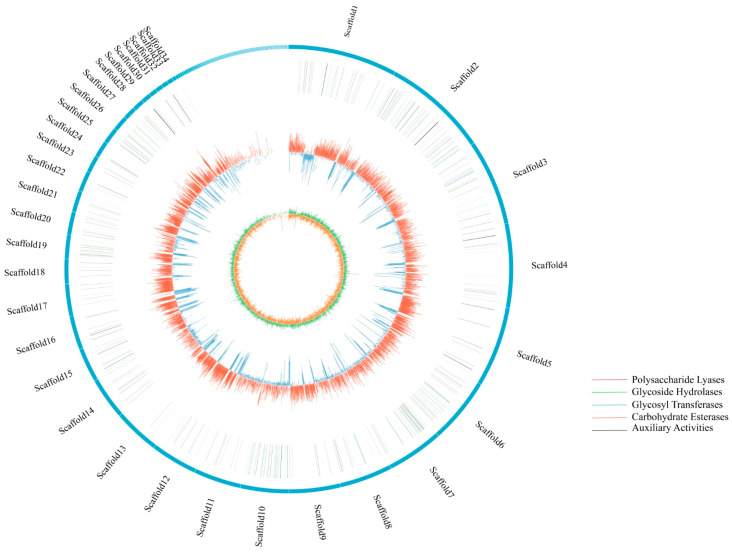
The circular graphical map of the whole *E. batatas* HD-1 genome.

**Figure 6 jof-10-00882-f006:**
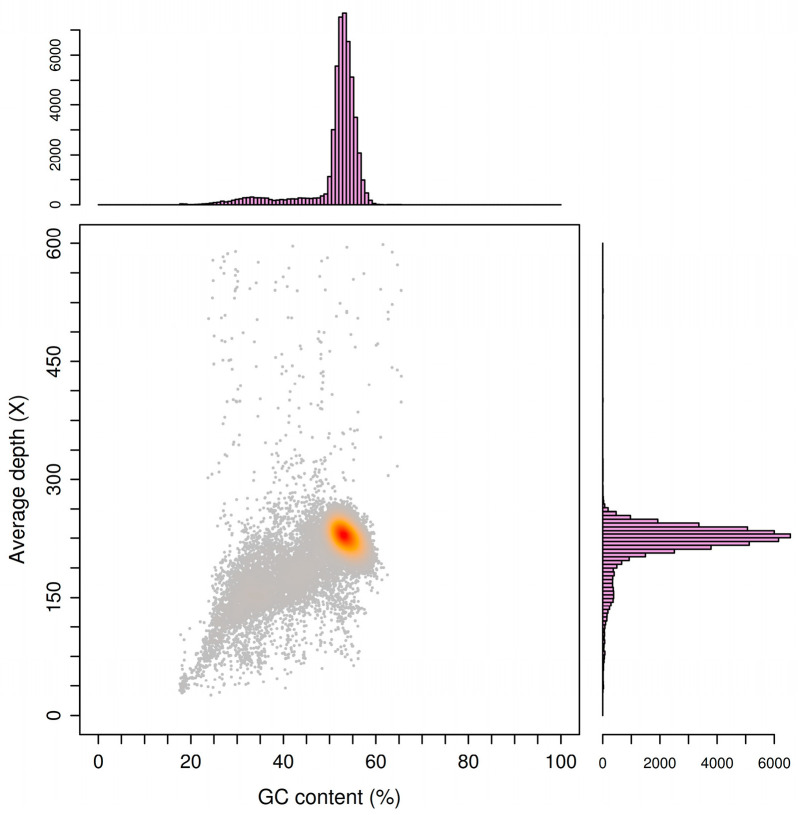
The GC_depth distribution diagram of *E. batatas* HD-1.

**Figure 7 jof-10-00882-f007:**
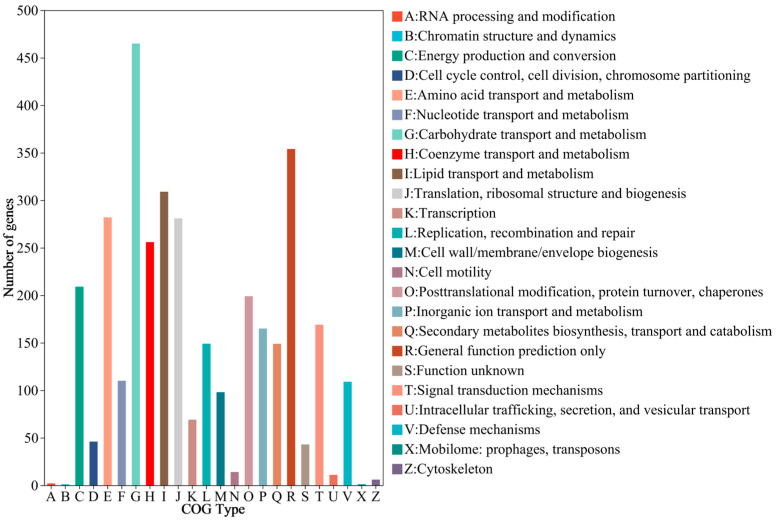
The COG function classification of HD-1 genes.

**Figure 8 jof-10-00882-f008:**
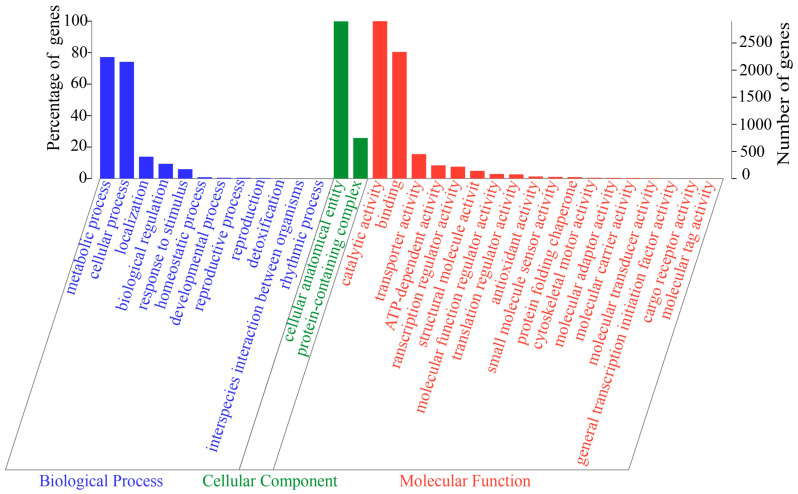
The GO function classification of HD-1 genes.

**Figure 9 jof-10-00882-f009:**
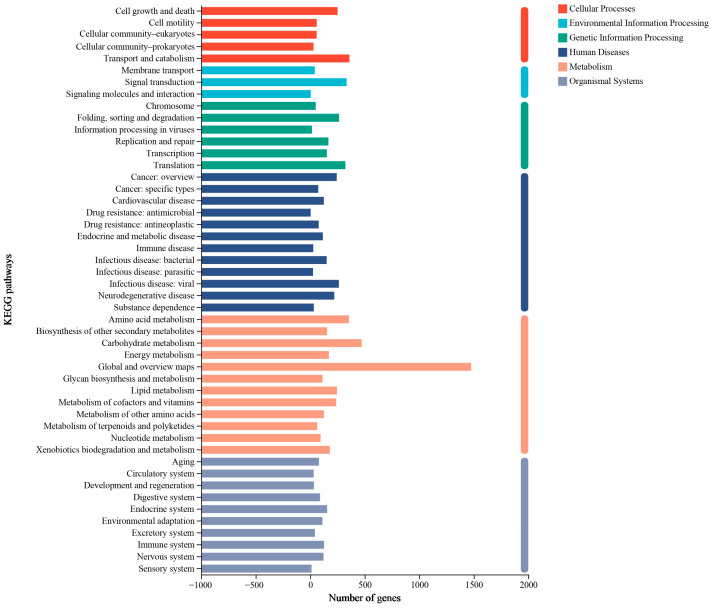
The KEGG function classification of HD-1 genes.

**Figure 10 jof-10-00882-f010:**
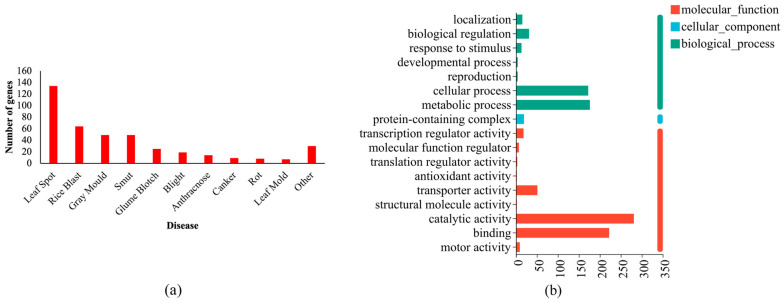
Analysis of genes encoding DFVF—Plant in the HD-1 genome: (**a**) genes predicted to encode DFVF—Plant. (**b**) GO functional annotation of genes encoding DFVF—Plant.

**Figure 11 jof-10-00882-f011:**
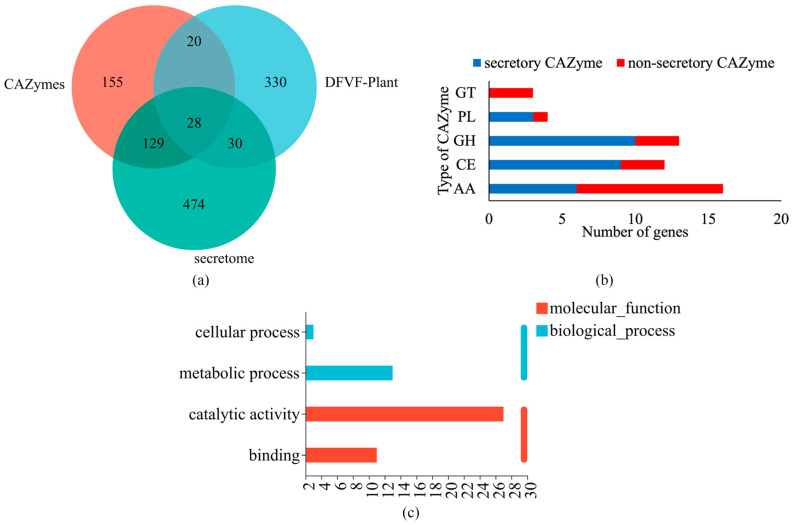
Analysis of genes encoding secretory CAZymes associated with plant diseases in the HD-1 genome: (**a**) Venn diagram of genes encoding DFVF—Plant, CAZymes, and secretome; (**b**) numbers of genes for different types of CAZymes involved in plant diseases; and (**c**) GO functional annotation of genes encoding secretory CAZymes associated with plant diseases.

**Figure 12 jof-10-00882-f012:**
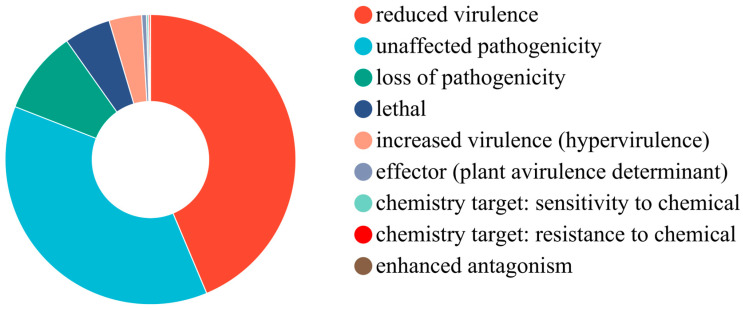
PHI functional classification of proteins encoded in the HD-1 genome.

**Figure 13 jof-10-00882-f013:**
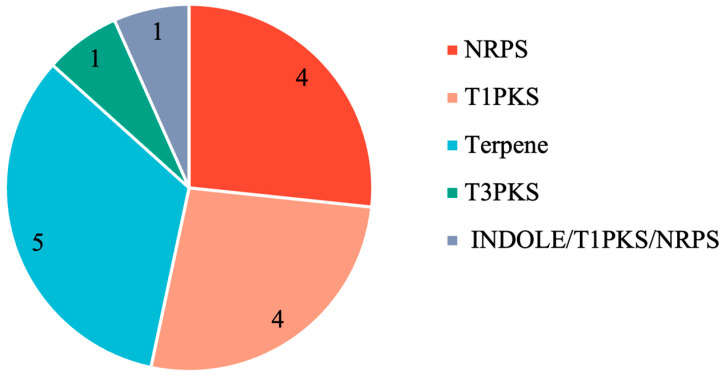
Analysis of secondary metabolite biosynthetic gene clusters.

**Figure 14 jof-10-00882-f014:**
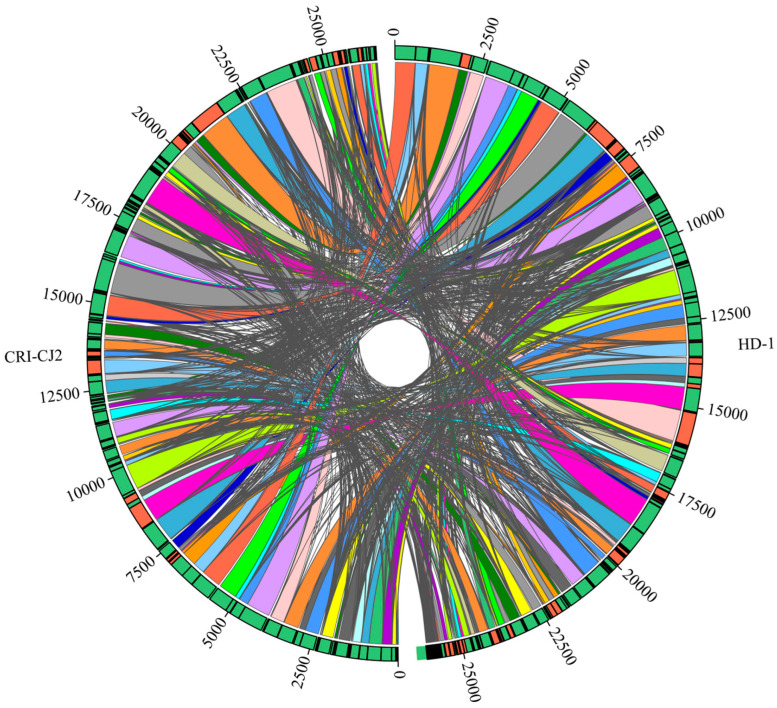
The synteny analyses between the *E. batatas* CRI-CJ2 and HD-1 genome. The colored area between two black dividing lines represents the collinear region aligned. Each connecting line represents a comparison record, where if it aligns positively, the outer ring colors of the comparison area are the same; if it aligns in reverse, the outer ring colors are different.

**Table 1 jof-10-00882-t001:** Scaffolds and genomic features of *E. batatas*.

Features	Value
Illumina sequencing (Gb)	6.04
PacBio_SMRT sequencing (Gb)	9.44
Scaffolds	167
Genome size (bps)	26,311,910
Largest sequence length (bps)	2,237,045
Least sequence length (bps)	1013
N50	1,049,739
N90	327,644
L50	9
L90	24
GC content (%)	50.81
Total genes number	7898
Total gene length (bps)	16,651,329
Average gene length (bps)	2108.3
Genes percentage of genome (%)	63.28
ncRNA	131
tRNA genes	150
5S rRNA	12
5.8S rRNA	3
18S rRNA	3
28S rRNA	8
Repeat sequences (bps)	3,995,791
LTRs	1153
SINEs	8
LINEs	230
DNA transposons	563

**Table 2 jof-10-00882-t002:** Statistical analysis of CAZy.

Type of CAZy	Number of Genes	Number of Families
GH	160	54
AA	59	13
CE	51	11
PL	12	5
GT	51	2

## Data Availability

This Whole-Genome Shotgun project has been deposited in GenBank under BioProject PR JNA1156576 and BioSample SAMN43494594.

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
