# Peer review of "Whole-Genome Sequencing and Genome Annotation of Pathogenic Elsinoë batatas Causing Stem and Foliage Scab Disease in Sweet Potato"

_jof, 2024, doi:10.3390/jof10120882_

Round 1

Reviewer 1 Report

The manuscript is devoted to the isolation and genome analysis of Elsinoë batatas, the sweet potato phytopathogen. During the analysis of the work, several comments arose.

Major:

I) The genome sequencing and assembly part is raw and lacks details, both in Methods and Results. For example:

1) How much data were obtained for the genome on the Illumina and PacBio platforms?

2) How did you use the Illumina data?

3) What tools and parameters were used for trimming?

4) How did you obtain scaffolds from the contigs? Looks like you got contigs rather than scaffolds.

5) Which dataset was used for BUSCO?

6) I could not find any PacBio data in the PRJNA1156576 BioProject.

7) “Additionally, 6.4% of the single-copy genes were not mapped, indicating a high level of genome completeness and assembly quality.

A confusing sentence.

8) “This assessment provides strong evidence that the genome assembly is of high quality and reliability, which is crucial for further downstream analyses and functional studies of E. batatas HD-1.

This statement has no support, at least a comparison with the available in NCBI E. batatas CRI-CJ2 genome should be performed – BUSCO and alignment. The E. batatas GCA_017309325.2 genome has an almost complete assembly level. It would even be possible to assemble your contigs into scaffolds if the strains are genetically close. The results of the comparison should be presented and discussed.

II) Some parts of the manuscript have significant similarities to the articles of other authors and should be rewritten. For example:

1) The abstract and some parts of the Results section are similar to the following manuscript – https://humgenomics.biomedcentral.com/articles/10.1186/s40246-023-00512-5.

2) The Illumina sequencing description was copied from the following manuscript – https://www.mdpi.com/2076-2607/11/12/2988.

Minor comments:

1) The title needs polishing. It is impossible to give a functional annotation to a strain, but it can be applied to its genome.

2) The genome size is given too precisely, it is impossible to eliminate all assembly errors. The genome size is usually given in Mb – 26.3 Mb in the case of Elsinoë batatas HD-1. The same is true for N50, etc. The base level resolution is too high, at least a kb level resolution should be used.

3) Abstract – “To comprehensively analyze the pathogenicity and proteomics of the isolated strain from a genetic perspective, whole-genome sequencing of E. batatas HD-1 was performed using both the PacBio and Illumina platforms.

Proteomics cannot be analyzed by WGS. Proteomics is a large-scale analysis of proteins.

4) It is also stated that the genome “contains 7,898 protein-coding genes, 131 noncoding RNAs, and 1,954 interspersed repetitive sequences”. The wording is incorrect, these data were predicted using bioinformatics tools even without RNA-Seq data. The term “predicted” should be used.

5) “For Illumina sequencing, genomic DNA from each strain was used for sequencing library construction.

From each strain? It appears that only one strain was analyzed in the manuscript.

6) Many reagents and instruments listed in the manuscript lack manufacturer information.

7) The font is too small in some figures.

Reviewer 2 Report

The manuscript “Whole-genome sequencing and functional annotation of pathogenic Elsinoë batatas causing stem and foliage scab disease in sweet potato” presents a genomic analysis of Elsinoë batatas, the pathogen responsible for scab disease in sweet potatoes, an important crop for food security in many regions. Although the genome of this pathogen has been sequenced before, the new sequencing of this isolate’s entire genome by the authors provides additional genomic data that supports the identification of specific virulence factors and enzymes involved in the infection process. This deeper insight into the pathogen’s genetic and infection mechanisms enhances understanding of this pathosystem and highlights potential tools for developing disease-resistant sweet potato varieties and effective control measures.

However, the study does not provide new evidence that can be directly applied. Additionally, the authors did not perform any comparative analysis between their genome annotation and the genome previously deposited in public databases.

Overall, the manuscript is well-written, and the data obtained is valuable to scientific communities involved in studies of plant-pathogen molecular interactions. 

Some corrections and suggestions have been provided throughout the manuscript.

Reviewer 3 Report

The manuscript titled "Whole-genome sequencing and functional annotation of pathogenic Elsinoë batatas causing stem and foliage scab disease in sweet potato" is devoted to a study of the pathogen  responsible for sweet potato stem and foliage scab disease.  Through a phylogenetic analysis based on the internal transcribed spacer (ITS) region of the ribosomal DNA, combined with morphological methods, the isolated strain was identified as Elsinoë batatas. To comprehensively analyze the pathogenicity and proteomics of the genome obtained for isolated strain E. batatas HD-1 was done using both PacBio and Illumina platforms. Functional analyses uncovered a diverse array of genes associated with pathogenicity, contributing to a better understanding of the molecular mechanisms underlying the infection process. The identification of many CAZy genes highlights the pathogen's ability to degrade plant cell walls, facilitating host infection. Additionally, 15 secondary metabolite gene clusters were identified, providing further insight into fungal virulence pathways and pathogenicity.. These findings suggest new potential targets for disease management strategies aimed at controlling sweet potato scab, ultimately contributing to improved agricultural practices and enhanced crop protection. This study enriches the genomic resources of E. batatas and provides a theoretical foundation for future investigations into the pathogenic mechanisms of its infection in sweet potatoes, as well as potential targets for disease control. The manuscript is well-written and will be of great interest to international research community.

It is clear that the manuscript has been considerably improved in its current version. The corresponding changes have been important for a better understanding of the text, and no further changes are necessary for the manuscript at this time.

Round 2

Reviewer 1 Report

Dear Authors,

Thank you for addressing most of my comments. However, several of them were not adequately addressed. Here they are:

a) Comments 2: How did you use the Illumina data?

Response 3: Thank you for your comment. The Illumina data is used for quality control. As described in lines 205-208, “The raw Illumina sequencing reads generated from the paired-end library were subjected to quality-filtered using fastp v0.23.0 with Q-score >=20, read length >=15, by which the low-quality data can be removed to form clean data”.

- I still do not understand how you used the Illumina data in your work. I understand that you filtered the data for quality and length, but what did you do next with the Illumina data?

b) Comments 4: How did you obtain scaffolds from the contigs? Looks like you got contigs rather than scaffolds.

Response 4: Thank you for your comment. The assembly software automatically performs error correction, trimming, and assembly according to overlap. By default, the minimum overlap length is 500.

- According to your description, these are contigs, not scaffolds. For example, see https://support.nlm.nih.gov/kbArticle/?pn=KA-03568.

c) Comments 6: I could not find any PacBio data in the PRJNA1156576 BioProject.

Response 6: Thank you for pointing this out. I have just uploaded the genome assembly file of PacBio data. you find related data according to the image below:

- I followed the images and it is Illumina data, not PacBio. Also, only 26.3 Mb are available in NCBI, while the manuscript says that 8.63 Gb of Illumina data were obtained. Actually it is the genome assembly, not the raw Illumina data as I thought before. So the raw Illumina data is missing as well as PacBio. The raw data should be available in the database. Moreover, the genome assembly should be deposited in NCBI Genome, not NCBI SRA, SRA is for raw data.

d) Comments 7: “Additionally, 6.4% of the single-copy genes were not mapped, indicating a high level of genome completeness and assembly quality.”

A confusing sentence.

Response 7: Thank you for pointing this out. 93.3% of the single-copy genes were completely aligned with the fungal reference set indicating a high level of genome completeness and assembly quality. So I rewrote it, as described in lines 312-315, “The result of BUSCO revealed that 93.3% of the single-copy genes were completely aligned with the fungal reference set, which indicates a high level of genome completeness and assembly quality”.

- 93.3% is not a high value, but perhaps it is normal not to have 6.7% of complete BUSCOs for Elsinoë batatas. The BUSCO completeness of the strain HD-1 genome assembly should be compared to the completeness of the strain CRI-CJ2 genome assembly to make such a conclusion.

e) Comments 12: The genome size is given too precisel, it is impossible to eliminate all assembly errors. The genome size is usually given in Mb – 26.3 Mb in the case of Elsinoë batatas HD-1. The same is true for N50, etc. The base level resolution is too high, at least a kb level resolution should be used.

Response 12: Thank you for pointing this out. I used “about 25.09 Mb” replaced “26,311,910 bp” in line 20. I used “- 25.09 Mb”, “- 1.00 Mb”, “- 0.31 Mb”, “- 15.88 Mb”, “- 3.8 Mb” replaced “26,311,910 bp”, “1,049,739”, “327,644”, “16,651,329”, “3,995,791” in lines 283-290.

- Why are the values in Mb smaller than in bp? For example, 26,311,910 bp is 26.31 Mb, not 25.09 Mb.

f) One new comment:

- Please, include the L50 and L90 values in Table 1.

See major comments

Reviewer 2 Report

The authors have addressed the questions raised, so the manuscript can be published. However, several figure captions need to be completed, as they must be self-explanatory.

Figure 2 (include medium and growth conditions)

Figure 3 (methodology)

Figure 4 (Inoculation and plant maintenance)

Figure 5-13 (methodology, namely bioinformatic tools used)

Table 2 (methodology, namely bioinformatic tools used)

Round 3

Reviewer 1 Report

Dear Authors,

Thank you for responding to my comments. However, it is disappointing to get some of the answers.

Mb – it is not megabits, it is megabases, 1 Mb = 1,000 kb = 1,000,000 b (bases).

Calculate L50 and L90 if you don't have them, use QUAST for example.

Illumina was a quality control for what? How did you do the quality control and what was the object of the quality control? Illumina data can indeed be used as a quality control, for example to assess the error rate in the obtained genome assembly, but I don’t see this information in the text.

I am also waiting for the raw PacBio and Illumina data to be available at NCBI.

See major comments
